# Effect of a mother-baby delivery pack on institutional deliveries: A community intervention trial to address maternal mortality in rural Zambia

Victor Mukonka[1,2], Cephas Sialubanje[2]*, Fionnuala M. McAuliffe[3], Olusegun Babaniyi[4], Sarai Malumo[4], Joseph Phiri[5], Patricia Fitzpatrick[6]

1 School of Medicine, Copperbelt University, Ndola, Zambia, 2 School of Public Health, Levy Mwanawasa Medical University, Lusaka, Zambia, 3 UCD Perinatal Research Centre, University College Dublin, National Maternity Hospital, Dublin, Ireland, 4 World Health Organization, Country Office, Lusaka, Zambia, 5 National Malaria Elimination Centre, Ministry of Health, Lusaka, Zambia, 6 School of Public Health, Physiotherapy & Sports Science, University College Dublin, Dublin, Ireland

* csialubanje@yahoo.com

**Data Availability Statement:** Data are available upon reasonable request from the TDRC ethics review board. This is based on the TDRC ethical

## Abstract

### Objectives

To test the effect of providing additional health education during antenatal care (ANC) and a mother-baby delivery pack on institutional deliveries in Monze, Zambia.

### Setting

16 primary health facilities conducting deliveries in the district.

### Participant

A total of 5000 pregnant women at any gestation and age attending antenatal care (ANC) services in selected health facilities were eligible for enrolment into the study. Out of these, 4,500 (90%) were enrolled into and completed the study. A total of 3,882 (77.6%) were included in the analysis; 12.4% were not included in the analysis due to incomplete data.

### Intervention

A three-year study (2012 to 2014) analysing baseline delivery data for 2012 and 2013 followed by a community intervention trial was conducted from January to December 2014. Health facilities on the western side were assigned to the intervention arm; those on the eastern side were in the control. In addition to the health education provided during routine ANC visits, participants in the intervention arm received health education and a mother-baby delivery pack when they arrived at the health facility for delivery. Participants in the control arm continued with routine ANC services.

policy. Address for the ethics committee: Tropical disease Research Centre (TDRC) Ethics Committee; Tel: +260212615444; email: tdrc-ethics@tdrc.org.zm. This restriction is according to the TDRC policy and indeed all other IRBs in Zambia. In addition, I wish to confirm that Dr Nawa Mukumbuta (mktnawa@gmail.com) and Ms Priscilla Funduluka (pfunduluka04@gmail.com), faculty members at Levy Mwanawasa Medical University, who did not collaborate in the study and not listed as authors on the manuscript, will separately and independently hold the de-identified data and respond to external requests for data access after clearance from the TDRC ethics review board according to the guidelines.

**Funding:** This work was supported by the World Health Organization, UNICEF and Zambian Ministry of Health (award/grant number: N/A) as part of the contribution to VM's doctoral research. UNICEF bought 1,700 mother-baby delivery packs and WHO contributed fuel for both delivery of the packs and trips for supervision. The Zambian Ministry of Health bought all mosquito nets for the packs and provided transport for distribution of packs to the health facilities in the intervention region. Please state what role the funders took in the study. If the funders had no role, please state: The funders had no role in study design, data collection and analysis, decision to publish, or preparation of the manuscript.

**Competing interests:** The authors have declared that no competing interests exist.

## Outcome measures

The primary measure was the number of institutional deliveries in both arms over the one-year period. Secondary measures were utilisation of ANC, post-natal care (PNC) and under-five clinic services. Descriptive statistics (frequencies, proportions, means and standard deviation) were computed to summarise participant characteristics. Chi-square and Independent T-tests were used to make comparisons between the two arms. One way analysis of variance (ANOVA) was used to test the effect of the intervention after one year (p-value<0.05). Analysis was conducted using R-studio statistical software version 4.2.1. The p-value<0.05 was considered significant.

## Results

Analysis showed a 15.9% increase in the number of institutional deliveries and a significant difference in the mean number of deliveries between intervention and control arms after one year (F(1,46) = 18.85, $p$<0.001). Post hoc analysis showed a significant difference in the mean number of deliveries between the intervention and control arms for 2014 (p<0.001). Compared to the control arm, participants in the intervention arm returned earlier for PNC clinic visit, brought their children back and started the under-five clinic visits earlier.

## Conclusion

These findings provide evidence for the effectiveness of the mother-baby delivery pack and additional health education sessions on increasing institutional deliveries, PNC and under-five children's clinic utilisation in rural Zambia.

## Trial registration

ISRCTN Registry (ISRCTN15439813 **DOI** 10.1186/ISRCTN15439813); Pan African Clinical Trial Registry (PACTR202212611709509).

## Introduction

### Background

Globally, approximately 830 women die every day from preventable causes related to pregnancy and childbirth [1, 2]. Most (99%) maternal deaths occur in developing countries, especially among women living in rural areas and among poorer communities [1, 2]. Zambia is one of the developing countries with a high maternal mortality ratio (MMR) [3, 4]—one of the highest in the world. The latest Demographic and Health Survey (DHS) [4] reported a high MMR of 252 per 100 000 live births in the country. Various factors including home deliveries and limited access to institutional deliveries have been shown to contribute to high MMR [5].

To improve maternal and newborn health outcomes, the Zambian Ministry of Health (MoH) has been implementing several life-saving interventions, including focused antenatal care (FANC) services [6, 7]. The national maternal health service guidelines recommend that pregnant women go for FANC visits at the health facility as soon as they realize they are pregnant and that they have at least four FANC visits [8]. The recommended FANC visit schedule prescribes that the first visit should occur by the end of 12 weeks of pregnancy, the second at

24 weeks, the third at 32 weeks, and the fourth at 36 weeks of pregnancy. However, women who experience discomfort, danger signs or have special needs or conditions beyond the scope of basic care may require additional visits [8]. Regular and prompt FANC is helpful to identify and prevent adverse pregnancy outcomes when it is sought early in pregnancy and is continued through delivery [9, 10].

FANC provides pregnant women with access to health messages and interventions for prevention and treatment of various diseases such as malaria in pregnancy, anaemia, and sexually transmitted infections (STIs) and HIV. The main services offered to pregnant women during FANC visits include HIV counselling, testing and treatment to prevent transmission of the virus to the unborn baby. Pregnant women are also tested and treated for syphilis, malaria, anaemia and hypertension in pregnancy. Moreover, women are offered health messages on nutrition, anaemia and birth preparedness. Other services include disease prevention against malaria and anaemia, through prophylaxis, micronutrient supplementation, immunization, provision of long-lasting insecticide nets (LLINs), and disease screening to ensure early detection of complications and prompt treatment. In addition, women are advised on birth preparedness and complication readiness [11].

Other interventions are delivery under supervision by a skilled birth attendant and postnatal care (PNC). The World Health Organisation (WHO) defines PNC as the care provided to the mother and her new-born child immediately after birth of the placenta and for the first six weeks (42 days) postpartum [12]. Prompt and quality PNC is a critical phase in the lives of mothers and newborns [13] for both the prevention and treatment of complications arising from pregnancy and delivery. PNC has the potential to avert a substantial proportion of maternal and perinatal mortality and morbidity. It also provides the mother with important information on family planning, well-baby, umbilical cord care, HIV and malaria prevention, and infant nutrition [14]. Since 2014, Zambia has been implementing the new PNC guidelines which seek to address the timing, number, and place of postnatal contacts, and content of PNC for all mothers and babies during the six weeks postpartum. After an uncomplicated vaginal delivery in a health facility by a skilled birth attendant, which is strongly recommended by the WHO, women and their new-borns are advised to remain within the health facility for a minimum of 24 hours for observation of danger signs, and prevention and treatment of postpartum complications, such as excessive bleeding, raised blood pressure or eclampsia. Women who give birth at home are encouraged to visit the health facility within 24 hours postpartum. Subsequent visits are at day 2–3, day 7–14, and day 42 postpartum [14]. This changed previous guidelines which required women to remain in the hospital for a shorter period of 6 h postpartum and return to the health facility after 6 days and 6 weeks.

Despite maternal healthcare services being provided at little to no cost in most government-run health facilities in Zambia, a large proportion of women still give birth at home. The ZDHS shows that one third (33%) of rural women in Zambia deliver at home without skilled birth attendants. Limited access and socioeconomic factors have been shown to be the main reasons influencing people's decision-making about utilisation of maternal healthcare services in developing countries, especially in rural areas [15]. Hidden costs incurred by expectant mothers when seeking maternal health care have been reported as important barriers to institutional deliveries. For instance, long distances and high poverty levels contribute to reduced access to health facilities. This is especially true for the 59% of Zambians living below the poverty datum line of one (1) US dollar per day [16]. Costs associated with seeking maternal healthcare include transportation, medications and supplies as well as the opportunity costs of travel and waiting time lost from productive activities [17]. It is common practice for health facilities to request expectant mothers to provide their own supplies for delivery. These supplies include clothing for both the mother and baby, baby blankets, napkins, baby soap and

delivery materials such as gloves and disinfectant [18]. Embarrassment at not being able to afford these basic requirements presents another relevant barrier to seeking professional care at health facilities. Out-of-pocket financing of health facility delivery costs has been shown to have substantial repercussions on households and make families more vulnerable to impoverishment [19]. This cost is prohibitive in rural populations that often rely on subsistence farming for their livelihood [20]. Reducing these hidden costs may motivate pregnant women to deliver in health facilities under the supervision of skilled birth attendants and contribute to reduction of maternal mortality [20]. Skilled care before, during and after childbirth can save the lives of women and newborn babies. The World Health Organisation (WHO) has shown that institutional delivery by skilled birth attendants and quality PNC are the most important strategies to reduce MMR in developing countries [21–24].

### Objectives

The primary objective of the study was to test the effect of providing additional health education during ANC and a mother-baby delivery pack on institutional deliveries in Monze, a rural district in Zambia. The mother-baby delivery pack included essential delivery supplies which would otherwise be an out-of-pocket expense for the mother. The secondary objective was to determine the effect of the intervention on ANC, PNC and under-five clinic utilisation by mother and baby. Evidence from the study is important for informing policy and interventions focusing on increasing institutional deliveries and improving maternal health outcomes.

## Materials and methods

### Trial design and randomisation

This was a three-year study (2012 to 2014) analysing baseline delivery data for 2012 and 2013 followed by one-year (1st January to 31st December, 2014) prospective community intervention trial conducted in Monze district, Zambia. Randomisation of study sites was done by the research team. First, the district was stratified into three (3) regions, namely, eastern, central and western regions. The central region was taken as a buffer zone separating the eastern and western regions. Next, by use of a coin flip, the western region was allocated to the intervention arm; the one on the eastern side as the control arm. A total of sixteen (16) health facilities were included in the study: eight (8) health facilities in each arm. A random sampling technique was used to select the 8 health centres from each arm. First, all the health centres in each arm were listed. The list served as a sampling frame. Next, information about delivery services at each health facility was obtained from the district health office. Health centres that did not conduct deliveries were removed from the list. A random number generator (RNG) was used to select the 8 health centres to be included in each arm. To be included, a health facility needed to have been conducting deliveries. The administrative centre of the district, with several urban health facilities, and two mission hospitals (Chikuni and Monze general) served as a physical separation and buffer between the intervention and control arms and prevented the spill over of relevant information on the intervention package. It was deemed unlikely for an expectant mother to by-pass the more than 30km buffer and go for delivery in the opposite arm. Both intervention and control regions had similar health facilities regarding the location (rural), size and catchment population, socio-economic and demographic profiles. The two regions mainly served peasant and subsistence farmers of the same tribe who shared the same cultural and traditional practices and beliefs. To ensure that the two regions were comparable regarding population size, population data for the two regions was obtained from the Central Statistical Office in Zambia [25].

## Study participants and setting

Study participants were pregnant women from both the intervention and control sites in Monze district in the Southern Province of Zambia. At the time, the district had a population of 203,038 [25] with 26 health centres and 2 mission hospitals (Chikuni and Monze general) under the catholic church. All the health centres (except for five) provided maternal and child health services and conducted deliveries. In addition to general health services, both Chikuni and Monze general mission hospitals provided routine and emergency obstetric and newborn care in the district. Monze mission general hospital served as the main referral hospital for the whole district, performed caesarean sections and dealt with all complicated cases referred from the health centres. There were no private health facilities providing obstetric and newborn health care in the district. At the time of the study, more than 50% of deliveries in the district took place at home, outside the health facility [25].

To be included in the study, women needed to be:

- Of reproductive age (15 years and above). Assent was obtained from the parents or legal guardians for the participants who were aged below 18 years

- Pregnant at any gestation

- Residing in the study site catchment area for at least 3 months. Pregnant women who were new in the area were excluded from the study

- Willing to participate

## Participant enrolment

Recruitment and enrolment of study participants was done by a pair of trained research assistants with support from the principal investigator. Study participants were identified and screened from the ANC clinics, when pregnant women went for their first ANC visit, regardless of the stage of their pregnancy (gestation). All pregnant women attending ANC from 1st January to 30th June 2014 were eligible to be enrolled into the study. Since the outcome of interest was place of delivery (institutional or home delivery), recruitment of study participants ended on 30th June 2014 to ensure that only women whose expected date of delivery (EDD) fell before 30th November 2014 were recruited into the study. This date would provide a one-month window for participants with prolonged gestation (beyond 40 weeks) to be followed up until their time of delivery, before the end of the study on 31st December 2014. It was expected that all participants would give birth within the follow up period and all the observations would be made; no participant was followed up after delivery.

## Intervention

The intervention comprised the mother-baby delivery pack and additional health education sessions. In addition to the health education provided during routine ANC visits, pregnant women in the intervention arm were provided with additional health education sessions. The additional health education was delivered by the health facility midwives through one-to-one and group discussion sessions. The sessions covered various aspects including birth preparedness, danger signs of pregnancy and complications. In order to ensure that standard and uniform information was provided during the health education sessions, leaflets translated into the local language were prepared by the research team.

In addition to the health education sessions, pregnant women in the intervention sites received a free mother-baby delivery pack. The packs were kept at the health facility; women only received them if they went to deliver at the health facility. Information about the mother-baby delivery pack was provided during the additional health education sessions which pregnant women had when they went for ANC visits. During the sessions, pregnant women were also given detailed information on the content of the delivery pack which included two (2) baby napkins, a bottle of Vaseline®, baby soap, a pair of delivery gloves, a baby vest, a chitenge (wrapper) and an insecticide treated mosquito net (ITN)). The control arm continued to receive routine standard ANC services.

## Outcomes

The primary outcome was the difference in the mean number of institutional deliveries between the intervention and control arms over a three-year period from 1st January 2012 to 31st December 2014. The secondary outcome measures were: 1) ANC service utilisation; 2) PNC service utilisation by mother and baby; 3) under-five clinic service utilisation. ANC, PNC and under-five service utilisation included the time of the first visit, number of visits completed according to the national guidelines. To measure ANC utilisation, the following information was collected: a) did the pregnant woman use ANC? b) what was the gestation at ANC booking? c) how many ANC visits did the pregnant woman make? Similarly, to operationalise PNC utilisation, the following information was collected: a) did the mother and her baby receive PNC? b) at what time after delivery were PNC services received? c) how many PNC visits did the mother-baby pair make? Similar questions were used to measure under-five clinic utilisation.

## Sample size estimation

From the district total population of 203, 038 in 2013 [25, 26], we estimated the expected deliveries for the district (using the United Nations Inter-Agency Group (WHO, UNICEF, UNFPA, World Bank) formula [27], which Zambia and other developing countries have adopted for estimating deliveries based on the population) to be 10,964. These would be pregnant and need maternal health services during 2014 when the intervention was implemented. Allowing for 60% of the population living in the urban area of the district, along the trial buffer zone, we estimated our sample size to be 4,500, (that is, 3000 in the intervention arm and 1,500 in the control).

## Data collection procedures

Data collection was done in two phases: baseline and intervention trial. To establish baseline delivery data prior to commencement of the intervention, year-long delivery records for 2012 and 2013 were reviewed and delivery data collected from the delivery registers at each health facility in the study sites, using a data extraction sheet. The data extraction sheet comprised various sections including demographics (age, place of residence), gravidity, parity, gestation, pre-existing medical conditions (hypertension, HIV, anaemia), expected date of delivery, date of delivery, place of delivery, mode of delivery, delivery outcome, baby condition and outcome. In addition, the PNC and under-five registers were reviewed to identify home deliveries. Next, the trial intervention data were collected from 1st January to 31st December 2014.

Both the baseline and intervention data were collected by a team of trained research assistants comprising a total of thirty-two (32) midwives working in pairs recruited from the health centres participating in the study. To avoid bias during data collection, several measures were taken. First, all the data collectors underwent a four (4) days training before commencement

of data collection. The training was conducted by the principal investigator assisted by a member of the research team and comprised classroom-based theory for three (3) three days followed by a practical training for one (1) day. Topics covered during classroom training included: a) the purpose of the study, b) study methods including sampling techniques and selection criteria; d) ethical considerations, and e) questionnaire administration. During the one day practical, data collectors practised how to use the checklist to collect baseline data, obtain informed consent and administer the questionnaire for the prospective study data. The consent form, checklist and questionnaire were revised based on the feedback from the data collectors. In addition, data collectors were not involved in the provision of ANC, delivery and PNC services. To be effective and efficient, data collectors worked in pairs under the supervision of the principal investigator.

Prospective data were collected using a paper-based questionnaire administered at three-time points: at enrolment into the study during the first ANC clinic visit; in the labour ward when the woman came for delivery, after delivery and when the woman and baby came for the first PNC visit. Sections in the questionnaire included demographic, socioeconomic and maternal history, ANC utilisation, place of delivery, date of delivery and outcomes, PNC and under five children's clinic utilisation.

## Statistical methods

Data from the checklist and questionnaire were entered into an Excel sheet and saved on a password-protected computer. After cleaning up, data was transferred into R-studio statistical software version 4.2.1 for analysis. Descriptive statistics (frequencies, percentages, proportions, means and standard deviations) were used to summarise participant socio-demographic and clinical data as well as institutional deliveries in intervention and control areas for each year (2012, 2013 and 2014). Before analysing the data for the primary outcome measure, the means and proportions between the two groups were compared using the independent T-test and Chi-square tests. After establishing that there was no significant difference in the baseline characteristics between the two groups, one way analysis of variance (ANOVA) was used to test the difference in the mean number of institutional deliveries between the intervention and control sites during the three years (2012 to 2014) under investigation. First, the main ANOVA was used to determine the overall effect of the intervention on institutional deliveries. Next, post hoc analysis using Bonferroni correction for pairwise comparison was conducted to determine the mean number of institutional deliveries between the intervention and control sites during the baseline period (2012 & 2013) and after introduction of the intervention (2014). Pairwise comparison was also conducted to measure the difference in the mean number of institutional deliveries in the intervention sites between the baseline period (2012 & 2013) and after the intervention. To determine the differences in ANC, PNC and under-five children clinic utilisation between the two groups, tests of significance (unpaired t-test and Chi-square) were computed. The p-value<0.05 was considered significant.

## Ethical statement

Ethical approval for the study was obtained from the tropical disease research centre (TDRC) ethics committee (TRC/ERC/04/09/2013). Authority to conduct the study was granted by the Ministry of Health, Zambia. Ethical exemption was obtained from University College Dublin, Ireland. Clinical trial registration was done with the ISRCTN registry **(ISRCTN15439813 DOI 10.1186/ISRCTN15439813)** and Pan African Clinical Trial Registry (**PACTR202212611709509**). In addition, we confirm that all methods were performed in accordance with the relevant guidelines regulating the consent to participate in a scientific

study. Before participants were enrolled into the study, midwives confirmed the pregnancy status using the rapid gravid index test. Next, the purpose of the study was explained to the women. To ensure informed consent, before data collection, participants were given and asked to read the consent form (S1 File) which was translated into the local language. Data collectors read the consent form for those who could not read. The consent form provided information on the background and purpose of the study and the data collection process. It also contained a detailed description of the participants' autonomy regarding their rights during the data collection process and the right to refuse or withdraw from the study. Potential benefits, risks and discomforts associated with the study were also explained. To make it easy for the participants to understand the consent form, research assistants explained every aspect in the language of the participants' choosing. To make it easy for the participants to understand the consent form, research assistants explained every aspect; they also requested the participants to ask questions or seek clarification, if they had any, before commencing data collection. Next, participants willing to participate in the study were asked to provide written informed consent; those who could not read or write were asked to mark with an 'X'. Individual informed consent was obtained from all the pregnant women during the first ANC visit before being enrolled into the study. After obtaining informed consent, the research assistants administered the questionnaire. In addition, in the case of minors, assent was obtained from guardians or parents. No further consent/assent was obtained at subsequent visits or data collection points.

## Patient and public involvement

Study participants and the public were not directly involved in the design of the study. Rather, the intervention was designed in response to the hidden barriers to institutional deliveries identified and expressed by the end users in the community during a cross-sectional study conducted prior to this community intervention trial. However, selection of the primary health facilities and study participants was done in collaboration with stakeholders from the national, provincial, district and primary health facility and community levels. First, pre-field meetings were held with the national, provincial and managers to select primary health care facilities to be included in the study. Next, local district managers selected the primary health facilities to be included in the study. In turn, primary health care facility leaders and midwives in consultation with the principal investigator, recruited and enrolled the participants into the study. Finally, a report was written and shared with the funding organisation, Ministry of Health for dissemination of study findings.

## Results

### Participants

A total of 5,000 pregnant women were identified as eligible for enrolment into the study. Of these, 4,500 (90%) pregnant women were enrolled into and completed the study, out of whom 618 (12.36%) had incomplete records and were not included in the final analysis. A total final sample of 3,882 respondents (77.6%) were included in the final analysis, comprising 2,684 (69.1%) from the intervention arm and 1,198 (30.9%) from the control arm. A summary of the recruitment algorithm of study participants is shown in the supporting information (S1 Fig).

### Demographic characteristics

Table 1 below shows the socio-demographic characteristics of the respondents from both the intervention and control arms. Most (83%) respondents were married with the majority (68.5%) having a gravidity of 1 to 3 at the time of recruitment. There was a significant

**Table 1. Demographic characteristics.**

| Variable | Intervention (n = 2,386 | Control (n = 1,198 | Total (n = 3,584 | P- value |
|---|---|---|---|---|
| Age (years) Mean (SD) | 24.6 (6.8) | 24.9 (6.6) | | **0.224** |
| Number of children Mean (SD) | 3.6 (2.3) | 3.4(2.2) | | **0.07** |
| **Marital status** | | | | **0.006** |
| Never married | 363 (15.2) | 105 (8.8) | 468 (13.1) | |
| Married | 1,952 (81.8) | 1,021 (85.2) | 2,973 (83.0) | |
| Cohabiting | 32 (1.3) | 42 (3.5) | 74 (2.1) | |
| Divorced | 21 (0.9) | 14 (1.2) | 35 (1.0) | |
| Separated | 11(0.5) | 11(0.9) | 22 (0.6) | |
| Widow | 7 (0.3) | 5 (0.4) | 12 (0.3) | |
| **Gravidity, n (%)** | | | | **0.001*** |
| 1 | 484 (20.3) | 341 (28.5) | 825 (23.0) | |
| 2–3 | 1,065 (44.6) | 567 (47.3) | 1,632 (45.5) | |
| 4–5 | 510 (21.4) | 201 (16.8) | 711 (19.8) | |
| +6 | 327 (13.7) | 89 (7.4) | 416 (11.6) | |
| Median (IQR) | 3.0 (2.0;4.0) | 2.0 (1.0;4.0) | 3.0 (2.0;4.0) | **0.001*** |
| **Antenatal booking (months)** mean (SD) | 5.4 (.2.3) | 5.5 (2.4) | | 0.21 |

difference in the median gravidity (p<0.001) between the respondents from the intervention arm (median = 3 pregnancies; IQR = 2.0, 4.0) and the control arm (median = 2 pregnancies; IQR = 1.0, 4.0). There was no significant difference ($p$ = 0.224) in the mean age between the mothers in the intervention (mean = 24.6 years, SD = 6.8) and control arms (mean = 24.9 years, SD = 6.6). There was a notable difference in the number of children between the respondents from the intervention (mean = 3.6, SD = 2.3) and control arms (mean = 3.4, SD = 2.2); however the difference did not reach statistical significance level (p = 0.07). There was no significant difference ($p$ = 0.21) in the mean gestational age at first ANC booking between the respondents in the intervention (mean = 5.4 months; SD = 2.3) and control arms (mean = 5.5 months; SD = 2.4).

## Primary outcome: Deliveries in interventions and control arms

Table 2 below shows the number of home and health facility deliveries in the intervention and control sites from January 2012 to December 2014. There was a 15.9% significant increase in

**Table 2. Deliveries in interventions and control arms.**

| Year | Intervention Sites | | | |
|---|---|---|---|---|
| | Intervention | Control | Total | P-value |
| 2012 | n = 4,074 | n = 2,814 | n = 6,888 | |
| Home | 2,394 (58.8) | 1,632 (58.0) | | >0.05 |
| Health Facility | 1,680 (41.2) | 1,182 (42.0) | | |
| 2013 | n = 4,154 | n = 2,950 | 7,104 | >0.05 |
| Home | 2,480 (59.7) | 1,628 (55.2) | | |
| Health Facility | 1,674 (40.3) | 1,322 (44.8) | | |
| 2014 | n = 4,197 | n = 2,990 | | <0.0001 |
| Home | 1,801 (42.9) | 1,776 (59.4) | | |
| Health Facility | 2,396 (57.1) | 1,214 (40.6) | | |

**Table 3. Pairwise comparisons using t tests with pooled SD (p-value adjustment method: Bonferroni).**

| Year/Arm | P values for pairwise comparison | | | | |
|---|---|---|---|---|---|
| | 2012 Intervention | 2012 Control | 2013 Intervention | 2013 Control | 2014 Intervention |
| 2012 Intervention | — | | | | |
| 2012 Control | 1.00 | — | | | |
| 2013 Intervention | 1.00 | 1.00 | — | | |
| 2013 Control | 0.56 | 1.00 | 0.16 | — | |
| 2014 Intervention | 0.048* | 0.0012** | 0.004* | 0.000066** | — |
| 2014 Control | 1.00 | 1.00 | 1.00 | 1.00 | 0.0017** |

the number of institutional deliveries in the intervention arm from 40.3% in 2013 to 57.1% (p< 0·0001) at the end of the intervention in December 2014. No such increase was seen in the control arm; the percentage of institutional deliveries dropped from 44.8% in 2013 to 40.6% at the end of 2014. The number of institutional deliveries in the control sites over the three years are shown in Table 2 below. The birth rates remained similar in both intervention and control regions over the three-year period 2012 to 2014.

## Analysis of variance (ANOVA) results

The main ANOVA showed a significant difference in the mean number of deliveries between the intervention and control arms at the end of the trial ($F(1,46) = 18.85$, $p<0.001$). Post hoc analysis showed a significant difference in the mean number of deliveries between the: a) intervention and control arms for 2014 ($p<0.001$); b) 2014 and the average baseline for 2012 and 2013 within the intervention arm ($p = 0.014$) and c) intervention arm in 2014 and control arm in 2013 ($p<0.001$). There was no significant difference between the intervention and control arms at baseline in 2012 and in 2013 ($p>0.05$). Table 3 below summarizes the ANOVA findings.

## Secondary outcome: ANC, PNC and under-five children's clinic utilisation

Table 4 shows comparison in maternal health service utilization between the intervention and control arms.

**ANC utilization.** We performed the Fisher's exact test to examine the association between ANC utilization and the intervention. The Fisher's exact test was 0.67 and not significant ($p>0.05$). The unpaired t-test showed a significant difference between the intervention and control groups regarding the time when women went for the first ANC visit. On average, women in the control arm (mean = 4.3 months, SD = 1.1) started their ANC visits earlier than those in the intervention arm (mean = 4.7, SD = 1.3), ($t(df = 3,482) = 9.13$, $p<0.001$)). The test also showed a significant difference in the mean number of ANC visits between the intervention (mean = 3.2 visits, SD = 0.8) and the control groups (mean = 3.3 visits (0.8), ($t(df = 3,482) = 3.53$, $p<0.001$)).

**PNC utilization.** The chi-square test of independence showed a significant association between the intervention and PNC utilization, $\chi^2(1, N = 3584) = 8.89$, $p = 0.03$. In addition, the unpaired t-test showed a significant difference between the intervention and control groups regarding the time when women returned for the first PNC visit. On average, women in the intervention arm (mean = 5.7 days, SD = 0.9) returned earlier for their first PNC visit than those in the control arm (mean = 7.8, SD = 4.0), ($t(df = 3,482) = 24.4$, $p<0.001$)).

**Table 4. Maternal health service utilisation.**

| Variable | Intervention (n = 2, 386) | Control (n = 1,198) | P-value |
|---|---|---|---|
|  | n (%)/ mean(sd) | n (%)/ mean(sd) |  |
| ANC attendance |  |  | 0.045 |
| Yes | 2,382 (99.8) | 1,197 (99.9) |  |
| No | 4 (0.2) | 1 (0.1) |  |
| First ANC booking (months) mean, sd | 4.7 (1.3) | 4.3 (1.1) | <0.001 |
| Number of ANC visits (mean (sd) | 3.2 (0.8) | 3.3 (0.8) | <0.001 |
| PNC |  |  |  |
| Yes | 2,344(98.2) | 1,158 (96.7) | 0.03 |
| no | 42 (1.8) | 40 (3.3) |  |
| Timing of PNC (days), mean (sd) | 5.7 (0.9) | 7.8 (4.0) | <0.001 |
| Under five children's clinic |  |  | <0.001 |
| Yes | 2, 158 (90.4) | 821 (68.5) |  |
| No | 228 (9.6) | 377 (31.5) |  |
| Timing of under five clinic attendance, mean (sd) | 30.4 (16.8) | 35.2 (14.1) | <0.001 |

**Under-five children's clinic utilization.** The chi-square test of independence showed a significant association between the intervention and under-five children's clinic utilization, $\chi^2$(1, N = 3584) = 272.95, $p$<0.0001. The unpaired t-test showed a significant difference between the intervention and control groups regarding the time when women brought their babies for the first under- five children's clinic visit. On average, women in the intervention arm (mean = 30.4 days, SD = 16.8) brought their babies earlier for their first under- five children than those in the control arm (mean = 35.2, SD = 14.1), (t(df = 3,482) = 8.5, p<0.001)).

## Discussion

The objective of this study was to test the effect of providing additional health education during ANC and a mother-baby delivery pack on institutional deliveries in Monze, Zambia. Overall, at the end of the intervention, our findings showed a significant 15.9% increase in the number of institutional deliveries in the intervention arm compared to the average at baseline. No such increase was seen in the control arm; the percentage of institutional deliveries dropped from 44.8% in 2013 to 40.6% at the end of 2014. ANOVA showed a significant difference in the mean number of deliveries between intervention and control arms at the end of the trial. Compared to women in the control arm, participants in the intervention arm returned early for PNC and under-five children's clinic visits.

These findings are consistent with previous studies which reported the importance of community-based interventions that focus on mitigating financial barriers in increasing facility-based deliveries [28–31]. A cluster randomised controlled trial measuring the impact of providing 'mama kit' incentives—small packages of childcare items provided to mothers conditional on delivering their baby in a facility in rural Zambia conducted by Wang and colleagues [31] reported a 63% statistically significant increase in facility delivery rates. Similarly, Akker and colleagues [32] reported a 78% increase in health facility deliveries following implementation of an intervention providing post-delivery packages consisting of one piece of soap, a baby blanket and a traditional wrap in Malawi. The study also reported that the increase was larger in peripheral rural facilities compared with the district hospital (94% vs. 38%) and concluded that mitigating financial barriers through provision of locally developed incentives can improve access to and utilization of professional maternity care in rural areas.

Moreover, our findings on the importance of financial barriers in preventing women from giving birth in health facilities are worth noting and corroborate previous studies [33–35]. For example, in their study on the reasons for home delivery and use of traditional birth attendants in rural Zambia, Sialubanje and colleagues [35] showed that, in addition to long distances, financial barriers including lack of money for baby clothes and other requirements prevent women from delivering at a clinic. These authors also showed that, although most pregnant women in rural Zambia have the intention to give birth in health facilities [36–38], many end up giving birth at home because they fail to provide the requirements asked by midwives at the health facility. To avoid being embarrassed, most pregnant women stay at home until delivery. They only go to the health facility for PNC and under- five children's clinic. Similarly, a systematic review [39] by Banke-Thomas and others reported on the importance of contextual factors in determining utilisation of facility-based delivery services. These findings confirm the importance of mitigating socioeconomic barriers in increasing health facility deliveries in through community interventions that provide incentive such as the mother-baby delivery pack.

Although interventions providing incentives to mitigate financial barriers to service utilization mainly focus on women's external motivation, previous studies have shown that such interventions have long term effects on women's continued utilization of facility delivery services including PNC and under-five children's clinic. For example, in their community health programme conducted in Narok County in Kenya, Kitui and colleagues [40] showed that such interventions had a significant impact on trends of facility deliveries over a 36-month period. Thus, such incentives have the potential to increase women's intrinsic motivation to use maternal health services in the long term, even in the absence of incentives, especially when the intervention is accompanied with health promotion activities. Health promotion has been shown to help increase utilisation of maternal health care services, reduce excess mortality, address the leading risk factors and underlying determinants of health, and strengthen sustainable health systems [41–44].

Finally, our findings show a significant difference between the intervention and control groups regarding utilisation of PNC and under five-children's clinics. Women in the intervention arm returned earlier for PNC and under five children's clinic compared to those in the control arm. These findings are consistent with previous studies which showed that women who give birth at health facilities are more likely to return to the health facility for their PNC, under-five children's clinic and subsequent deliveries [14, 45–49]. Thus, the findings highlight the effect of the intervention on women's motivation to continuously use maternal health services after delivery. Going to the health facility for intra-partum care allows the woman and her new-born baby to receive care from a skilled birth attendant [50]. They compare the advantages of giving birth at the health facility with the dangers of delivering at home without skilled care. Moreover, during these visits, women receive health promotion messages about their health and that of the baby. These benefits motivate them to return to the health facilities in future. These findings highlight the importance of mitigating financial and other barriers in ensuring continuum of care for women and their newborn babies. They also suggest that mitigating barriers to institutional deliveries could motivate women to continue utilisation of other maternal and newborn health services even in the absence of an incentive such as the mother-baby pack.

## Limitations

Possible limitations for the study should be acknowledged. First, use of one rural district in the southern part of the country may limit external validity of the findings to other regions with

different geographical and demographic characteristics. Second, non-random allocation of facilities could have introduced selection bias. Further, the intervention was conducted over a one-year period; the long-term effects of the intervention on facility deliveries are not clear; follow-up studies are required to evaluate the impact of the intervention on women's motivation to continuously use health facility delivery services after the intervention. In addition, we could not run analysis of covariance (ANCOVA) to account for the differences in gravidity. This could have led to confounding in our findings. However, most baseline variables were comparable between the intervention and control health facilities. In addition, the study did not measure the impact of the intervention on maternal mortality as it was out of its scope; evaluation studies are required to confirm this relationship. The study did not collect data on the healthcare related costs of the intervention. A follow-up study is required to determine the full cost of implementing such an intervention nationally.

Despite these limitations, we believe this study has brought out importance evidence on the effect of provision of a non-monetary incentive on increasing institutional deliveries, PNC and under-five children's clinic. In addition, community intervention trials are the only appropriate study design suited for the evaluation of lifestyle interventions that cannot be allocated to individuals [51–54]. As opposed to clinical trials, community intervention trials focus on public health interventions that are focused on reducing the risk or burden of disease and/or mortality within a community context. Our use of this design employing a large sample size increased the external validity of the findings. The effective buffer area between the intervention and control regions prevented women from crossing over to either side, minimised the spillover effect and potential misclassification bias. To our knowledge, this is the first study to be undertaken to evaluate effects of provision of non-financial incentives in increasing institutional deliveries in rural Zambia.

## Conclusion

These findings provide evidence for the effectiveness of the additional health education sessions and a mother-baby delivery pack on increasing institutional deliveries, PNC and under-five children's clinic utilisation in rural Zambia. They could serve as a basis for policies and interventions focusing on designing effective interventions to address reversible barriers to health facility delivery, PNC and under-five children's clinic under skilled care, especially in developing countries where poverty, home deliveries and high maternal mortality are high. Future studies should focus on 1) measuring the impact of the intervention on maternal mortality reduction and improving maternal health outcomes, and 2) determining the cost-effectiveness of such an intervention.

## Supporting information

**S1 Fig. Participant recruitment algorithm.**
(PDF)

**S1 Checklist. CONSORT 2010 checklist of information to include when reporting a randomised trial.**
(PDF)

**S1 File.**
(PDF)

**S2 File.**
(PDF)

**S3 File.**
(PDF)

**S4 File.**
(PDF)

## Author Contributions

**Conceptualization:** Victor Mukonka, Cephas Sialubanje.

**Data curation:** Victor Mukonka.

**Formal analysis:** Victor Mukonka.

**Funding acquisition:** Victor Mukonka.

**Investigation:** Victor Mukonka.

**Methodology:** Victor Mukonka, Cephas Sialubanje, Patricia Fitzpatrick.

**Project administration:** Victor Mukonka.

**Resources:** Victor Mukonka.

**Software:** Victor Mukonka.

**Supervision:** Victor Mukonka.

**Validation:** Victor Mukonka.

**Visualization:** Victor Mukonka.

**Writing – original draft:** Victor Mukonka, Cephas Sialubanje.

**Writing – review & editing:** Victor Mukonka, Cephas Sialubanje, Fionnuala M. McAuliffe, Olusegun Babaniyi, Sarai Malumo, Joseph Phiri, Patricia Fitzpatrick.

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
