## [Decision Letter · Decision Letter 0]

11 Apr 2023

PONE-D-23-04893Effect of a non-financial incentive on institutional deliveries: A community intervention trial to address maternal mortality in rural ZambiaPLOS ONE

Dear Dr. Sialubanje,

Thank you for submitting your manuscript to PLOS ONE. After careful consideration, we feel that it has merit but does not fully meet PLOS ONE’s publication criteria as it currently stands. Therefore, we invite you to submit a revised version of the manuscript that addresses the points raised during the review process.

We look forward to receiving your revised manuscript.

Kind regards,

G. K. Balasubramani

Academic Editor

PLOS ONE

Journal Requirements:

3. Please include the following request in the decision letter, and ping me with follow up. “Please include a complete copy of PLOS’ questionnaire on inclusivity in global research in your revised manuscript. Our policy for research in this area aims to improve transparency in the reporting of research performed outside of researchers’ own country or community. The policy applies to researchers who have travelled to a different country to conduct research, research with Indigenous populations or their lands, and research on cultural artefacts. The questionnaire can also be requested at the journal’s discretion for any other submissions, even if these conditions are not met.  Please find more information on the policy and a link to download a blank copy of the questionnaire here: https://journals.plos.org/plosone/s/best-practices-in-research-reporting. Please upload a completed version of your questionnaire as Supporting Information when you resubmit your manuscript.

4. Please ensure you have included the registration number for the clinical trial referenced in the manuscript.

"This work was supported by the World Health Organization, UNICEF and Zambian Ministry of Health (award/grant number: N/A) as part of the contribution to VM's doctoral research.  UNICEF bought 1,700 mother-baby delivery packs and WHO contributed fuel for both delivery of the packs and trips for supervision. The Zambian Ministry of Health bought all mosquito nets for the packs and provided transport for distribution of packs to the health facilities in the intervention region."

6. Thank you for stating the following in your Competing Interests section:  

'No authors have competing interest"

7. We note that you have indicated that data from this study are available upon request. PLOS only allows data to be available upon request if there are legal or ethical restrictions on sharing data publicly. For information on unacceptable data access restrictions, please see http://journals.plos.org/plosone/s/data-availability#loc-unacceptable-data-access-restrictions. 

8. Please amend either the title on the online submission form (via Edit Submission) or the title in the manuscript so that they are identical.

9. Please amend the manuscript submission data (via Edit Submission) to include authors Dr. Kangwa I. M. Muma, Dr. Robin Bailey, Dr. Jessie I. M. Nyalazi, Dr. George Zulu, Dr. Tyness S. Mumba-Malisawa, Dr. Lillian M. L. Chinama-Musonda, Dr. Kachikonyo Sibande-Muma, Dr. Consity Mwale, Dr. Alex Makupe, Dr. Gardner Syakantu, Dr. Patrick Kaonga, Dr. Edgar Simulundu and Dr. Charles Michelo.

10. Please amend your authorship list in your manuscript file to include author Dr. Victor Mukonka, Dr. Fionnuala M McAuliffe, Dr. Olusegun Babaniyi, Dr. Sarai Malumo, Dr. Joseph Phiri and Dr. Patricia Fitzpatrick.

11. Please amend either the abstract on the online submission form (via Edit Submission) or the abstract in the manuscript so that they are identical.

12. Please include your full ethics statement in the ‘Methods’ section of your manuscript file. In your statement, please include the full name of the IRB or ethics committee who approved or waived your study, as well as whether or not you obtained informed written or verbal consent. If consent was waived for your study, please include this information in your statement as well. 

13. Please include captions for your Supporting Information files at the end of your manuscript, and update any in-text citations to match accordingly. Please see our Supporting Information guidelines for more information: http://journals.plos.org/plosone/s/supporting-information. 

Reviewers' comments:

Reviewer's Responses to Questions

**Comments to the Author**

1. Is the manuscript technically sound, and do the data support the conclusions?

Reviewer #1: Yes

2. Has the statistical analysis been performed appropriately and rigorously? 

Reviewer #1: Yes

3. Have the authors made all data underlying the findings in their manuscript fully available?

Reviewer #1: Yes

4. Is the manuscript presented in an intelligible fashion and written in standard English?

Reviewer #1: Yes

5. Review Comments to the Author

Reviewer #1: A matched case-control clinical trial was conducted which aimed to investigate the association of myocilin gene mutations in Primary Open Angle Glaucoma (POAG). MYOC mutations were significantly associated with POAG. In fact, patients with MYOC mutations were 1.8 times more likely to develop POAG compared to those without the mutation.

Minor revisions:

1- Abstract: Name the statistical method used for the bivariate analysis.

2- Statistical Methods section:

a. Identify the descriptive statistical method used.

b. List the confounders used in the multivariate regression analysis.

c. Cite the statistical software used for the analysis.

d. State the criteria for determining a statistically significant p-value. For example include a statement similar to the following. “P-values less than 0.05 were considered statistically significant.”

3- Table 1: Since the study was not randomized, provide p-values to compare socio-demographic factors between the two groups.

4- Page 11: Name the statistical method used for the bivariate analysis.

5- Table 5: The title of table 5 is misleading since both univariate and multivariate logistic regression results are presented.

6- To assist in the review process, add line numbering to the document.

6. PLOS authors have the option to publish the peer review history of their article (what does this mean?). If published, this will include your full peer review and any attached files.

Reviewer #1: No

---

## [Author Response · Author response to Decision Letter 0]

8 May 2023

Editorial comments

1. Please ensure that your manuscript meets PLOS ONE's style requirements, including those for file naming. The PLOS ONE style templates can be found at:

and https://journals.plos.org/plosone/s/file?id=ba62/PLOSOne_formatting_sample_title_authors_affiliations.pd,

Response: We have formatted the manuscripts according to the PLOS ONE style

Response: We have edited this section accordingly (see page 8)

3. Please include the following request in the decision letter and ping me with follow up. “Please include a complete copy of PLOS’ questionnaire on inclusivity in global research in your revised manuscript. Our policy for research in this area aims to improve transparency in the reporting of research performed outside of researchers’ own country or community. The policy applies to researchers who have travelled to a different country to conduct research, research with Indigenous populations or their lands, and research on cultural artefacts. The questionnaire can also be requested at the journal’s discretion for any other submissions, even if these conditions are not met. Please find more information on the policy and a link to download a blank copy of the questionnaire here: https://journals.plos.org/plosone/s/best-practices-in-research-reporting. Please upload a completed version of your questionnaire as Supporting Information when you resubmit your manuscript.

Response: We have downloaded the questionnaire, filled it in and uploaded it.

4. Please ensure you have included the registration number for the clinical trial referenced in the manuscript.

Response: The registration numbers for referenced clinical trials have been included (see reference# 14 and 16)

5. Thank you for stating the following financial disclosure: "This work was supported by the World Health Organization, UNICEF and Zambian Ministry of Health (award/grant number: N/A) as part of the contribution to VM's doctoral research. UNICEF bought 1,700 mother-baby delivery packs and WHO contributed fuel for both delivery of the packs and trips for supervision. The Zambian Ministry of Health bought all mosquito nets for the packs and provided transport for distribution of packs to the health facilities in the intervention region.

Response: We appreciate this guide. We have now edited this section to state that the: “The funders had no role in study design, data collection and analysis, decision to publish, or preparation of the manuscript." (see page 13)

6. Thank you for stating the following in your Competing Interests section: 

'No authors have competing interest". Please complete your Competing Interests on the online submission form to state any Competing Interests. If you have no competing interests, please state "The authors have declared that no competing interests exist.", as detailed online in our guide for authors at http://journals.plos.org/plosone/s/submit-now. This information should be included in your cover letter; we will change the online submission form on your behalf

Response: We appreciate the guidance. We have now edited this part in the manuscript to read: “The authors have declared that no competing interests exist.” (see page 13)

7. We note that you have indicated that data from this study are available upon request. PLOS only allows data to be available upon request if there are legal or ethical restrictions on sharing data publicly. For information on unacceptable data access restrictions, please see http://journals.plos.org/plosone/s/data-availability#loc-unacceptable-data-access-restrictions

7a) If there are ethical or legal restrictions on sharing a de-identified data set, please explain them in detail (e.g., data contain potentially identifying or sensitive patient information) and who has imposed them (e.g., an ethics committee). Please also provide contact information for a data access committee, ethics committee, or other institutional body to which data requests may be sent

Response: We appreciate this comment. There are no legal restrictions on deidentified data as such. However, it is a TDRC ethics committee policy, just like many other IRBs in Zambia and elsewhere, that express permission should be sought from and granted by the ethics committee before data are publicly shared. The principal investigator can request for this permission if the journal needs the data. The corresponding author can then share it. We have now edited this section to read: Data are available upon reasonable request from the corresponding author and with permission of the TDRC ethics review board. This is based on the TDRC ethical policy (see page 13) 

Address for the ethics committee: Tropical disease Research Centre (TDRC) Ethics Committee; Tel: +260212615444; email:tdrc-ethics@tdrc.org.zm

b) If there are no restrictions, please upload the minimal anonymized data set necessary to replicate your study findings as either Supporting Information files or to a stable, public repository and provide us with the relevant URLs, DOIs, or accession numbers. Please see http://www.bmj.com/content/340/bmj.c181.long for guidelines on how to de-identify and prepare clinical data for publication. For a list of acceptable repositories, please see http://journals.plos.org/plosone/s/data-availability#loc-recommended-repositories

Response: This explained in section “a” above

8. Please amend either the title on the online submission form (via Edit Submission) or the title in the manuscript so that they are identical

Response: The title has been amended accordingly

9. Please amend the manuscript submission data (via Edit Submission) to include authors Dr. Kangwa I. M. Muma, Dr. Robin Bailey, Dr. Jessie I. M. Nyalazi, Dr. George Zulu, Dr. Tyness S. Mumba-Malisawa, Dr. Lillian M. L. Chinama-Musonda, Dr. Kachikonyo Sibande-Muma, Dr. Consity Mwale, Dr. Alex Makupe, Dr. Gardner Syakantu, Dr. Patrick Kaonga, Dr. Edgar Simulundu and Dr. Charles Michelo

 Response: These authors are not part of this study; they are appearing on the other manuscript we submitted to this journal. The two submissions are different. These authors are on the study: Association between synonymous myocilin mutations and primary open angle glaucoma: A case-control study of patients attending selected referral eye care health facilities in Zambia

10. Please amend your authorship list in your manuscript file to include author Dr. Victor Mukonka, Dr. Fionnuala M McAuliffe, Dr. Olusegun Babaniyi, Dr. Sarai Malumo, Dr. Joseph Phiri and Dr. Patricia Fitzpatrick

Response: These authors are already appearing on the manuscript (see page 1)

11. Please amend either the abstract on the online submission form (via Edit Submission) or the abstract in the manuscript so that they are identical.

Response: This has been done accordingly and the two are now identical (see page 2)

12. Please include your full ethics statement in the ‘Methods’ section of your manuscript file. In your statement, please include the full name of the IRB or ethics committee who approved or waived your study, as well as whether or not you obtained informed written or verbal consent. If consent was waived for your study, please include this information in your statement as well.

Response: The ethics statement in the methods section has been edited to include a full statement (see page 8)

13. Please include captions for your Supporting Information files at the end of your manuscript, and update any in-text citations to match accordingly. Please see our Supporting Information guidelines for more information: http://journals.plos.org/plosone/s/supporting-information

Response: This has been done (see page 14)

Response: The reference list has been reviewed and updated accordingly. No retracted articles have been cited.

Reviewer's Responses to Questions

1. Abstract: Abstract: Name the statistical method used for the bivariate analysis

Response: We have edited the abstract and indicated that we used the independent T-test and Chi-Square test to measure the differences in means and proportions between the two groups (see page 2) 

2. Statistical Methods section:

a. Identify the descriptive statistical method used.

b. List the confounders used in the multivariate regression analysis.

c. Cite the statistical software used for the analysis.

d. State the criteria for determining a statistically significant p-value. For example include a statement similar to the following. “P-values less than 0.05 were considered statistically significant.”

Response: We appreciate the comment; We have included this information in the abstract and mentioned that analysis were conducted using using R-studio statistical software version 4.2.1 (see page 2).

3. Table 1: Since the study was not randomized, provide p-values to compare socio-demographic factors between the two groups

Response: We appreciate the guidance; the p-values have been provided in table 1.

4. Name the statistical method used for the bivariate analysis

Response: We have edited the abstract and indicated that we used the independent T-test and Chi-Square test to measure the differences in means and proportions between intervention and control groups (see page 7)

5. Table 5: The title of table 5 is misleading since both univariate and multivariate logistic regression results are presented

Response: We are not sure about this comment; we only submitted four tables and we did not carry out regression analysis. We conducted ANOVA which we presented in table 4. It appears there was a mixup on the two studies we submitted.

6. To assist in the review process, add line numbering to the document.

Response: We have added line numbering accordingly.

---

## [Decision Letter · Decision Letter 1]

10 Aug 2023

PONE-D-23-04893R1Effect of a non-financial incentive on institutional deliveries: A community intervention trial to address maternal mortality in rural ZambiaPLOS ONE

Dear Dr. Sialubanje,

Thank you for submitting your manuscript to PLOS ONE. After careful consideration, we feel that it has merit but does not fully meet PLOS ONE’s publication criteria as it currently stands. Therefore, we invite you to submit a revised version of the manuscript that addresses the points raised during the review process.

The manuscript has been evaluated by three reviewers, and their comments are available below. The reviewers have raised a number of major concerns. In particular, they request major revisions to improve the quality of the reporting throughout the manuscript. There are also concerns about the quality of the analyses, and the conclusions should be revised to ensure that they are presented appropriately.

Could you please carefully revise the manuscript to address all comments raised?

We look forward to receiving your revised manuscript.

Kind regards,

Marianne Clemence

Staff Editor

PLOS ONE

Reviewers' comments:

Reviewer's Responses to Questions

**Comments to the Author**

1. If the authors have adequately addressed your comments raised in a previous round of review and you feel that this manuscript is now acceptable for publication, you may indicate that here to bypass the “Comments to the Author” section, enter your conflict of interest statement in the “Confidential to Editor” section, and submit your "Accept" recommendation.

Reviewer #1: (No Response)

Reviewer #2: (No Response)

Reviewer #3: (No Response)

2. Is the manuscript technically sound, and do the data support the conclusions?

Reviewer #1: Yes

Reviewer #2: No

Reviewer #3: Partly

3. Has the statistical analysis been performed appropriately and rigorously? 

Reviewer #1: Yes

Reviewer #2: No

Reviewer #3: Yes

4. Have the authors made all data underlying the findings in their manuscript fully available?

Reviewer #1: No

Reviewer #2: Yes

Reviewer #3: No

5. Is the manuscript presented in an intelligible fashion and written in standard English?

Reviewer #1: Yes

Reviewer #2: Yes

Reviewer #3: No

6. Review Comments to the Author

Reviewer #1: Minor revision:

1- Line 61: R-studio is a development environment for R. It is important to state the version of R that was used for conducting the analysis. In fact, the use of R-studio is rarely noted in manuscripts.

2- Line 59: Chi-square and independent t-tests are used to make comparisons between arms. Be sure to include the word "compare" or "comparisons" in this sentence.

3- Line 61: Clarify if "(p-value < 0.05)" implies that p-values < 0.05 were considered statistically significant.

4- Line 198: The independent t-test and chi-square tests are actually inferential tests rather than descriptive statistics. Descriptive statistics are means, standard deviations, frequencies, and percentages, etc.

5- Supplementary Table 1 contains replicates of the variables.

6- Supplementary Tables 2 and 3: State a more precise p-values rather than > 0.05.

Note: Line numbers refers to those in the tracked changes of revision 1.

Reviewer #2: General:

- The introduction is too brief and lacks key information for the reader. There is no information on ANC or on related interventions and their effectiveness. What have we already learned that led the authors to design and test this specific intervention? There should be sufficient information that a reader can glean this from the introduction.

- This manuscript reports findings from a randomized trial. The CONSORT guidelines should be used to ensure all relevant information is included. I suggest the authors download the CONSORT checklist and verify all information is included.

- The statistical analysis is not appropriate for the trial design. This was a cluster randomized trial; therefore, facility-level clustering needs to be accounted for in the analysis or standard errors will not be approrpriately estimated, potentially biasing the results and leading to incorrect interpretation. I have provided more details below. Moreover, characteristics that were different between arms at baseline should be adjusted for.

- There should be some analysis of participants lost to follow up. Are these women significantly different? How might this bias results?

- I cannot fully evaluate the results or discussion at this time given the limitations of the methods (lack of information).

- Were women not exposed to the intervention at different points of their pregnancy? The authors should expand on the timing of outcome measures relative to the intevention. If women were enrolled at their first ANC visit, it is unclear to me whether they were observed after the conclusion of the intervention as this is not clear from the methods section.

Abstract:

- Please define secondary outcomes reported as results here.

- The abstract attributes the difference in arms to the mother-baby pack, but the abstract notes that the intervention also included health education. Is the difference attributed to both or can the authors statistically isolate the effect of the mother-baby pack? Either way, this should be clear in the abstract.

Introduction:

- Line 82-83: “Home deliveries…” this sentence makes a claim that should be supported by a citation(s).

- Why is there no information or literature review on ANC in the introduction if this is a key component of the intervention evaluated? Please describe relevant background on ANC, including number of recommended visits and attendance rates (e.g., from the most recent DHS) in Zambia, and barriers to ANC (could be more broadly, as relevant to the study populatoin).

Methods:

- Are there any private facilities that can support deliveries?

- The secondary outcome measures need to be more clearly defined. For example, “knowledge of pregnancy danger signs” is vague.

- How was the randomization conducted?

- Inofrmation about recruitment should be in the enrollment section (around line 167).

- The abstract states analysis was done in R but the methods section states analysis was conducted in SPSS. Please clarify so that this is consistent.

- The inforamtion about data collection is insufficient. Please state where 2012 and 2013 data come from, and exactly what data was collected (e.g., precisely what measures were on the data extraction sheet, who extracted data). What is the timing of paper-based questionnaires (“PNC” is far too vague, for example). How were interviewers trained? Where were interviews conducted?

- All non-descriptive analyses require clustering of the standard errors by facility given the study design. The authors should use regression models (e.g., linear, logistic depending on the outcome) with standard errors clustered by health facility. Otherwise, the independence assumption is violated which will bias standard errors and thus potentially lead to incorrect inferences about the statistical significance of the estimated relationships.

- Regression models estimating the effect of the intervention should also adjust for factors that were significantly differnet between arms at baseline, such as gravidity. Otherwise, this may be a spurious relationship.

Results:

- Please include some analysis of participants lost to follow up. Are these women different from retained participants, and if so, how?

Limitations:

- The authors should describe biases related to non-random allocation of facilities and whether there are potential selection biases related to including only public-sector facilities.

Minor comments:

- Line 84: Demographic & Health Survey should be capitalized

- Line 108: chane aim to objective, remove “study also” for clarity

- Lines 252 and 254: please include units

Reviewer #3: This is a very important and informative study. However, I have a few concerns that need to be addressed. The statement "Participants in the intervention arm received ANC,health education and a mother-baby delivery pack when they arrived at the health facility for delivery" in your abstract is misleading. Please re-write.

Please re-write objectives stated in line 108-109 to align with the abstract and the rest of the document. The document still has alot of typos; e.g line 326-importance versus important; 327 "community intervention trials are the only appropriate study design suited...". Please revisit and re-write for clarity.

You mention that study participants were identified from the first ANC visits, was this part of your screening criteria?

You later state these differences in healthservice utilization, alongside other variables like gravidity, first ANC utilization, how did you take care of these and other obvious confounders? Please describe your multivariate analysis in more detail.

Please report ALL secondary outcomes in the outcome tables and results section before embarking on their discussion.

Line 257 states, "health facility deliveries in the intervention and control sites from January 2012 to Dec 2014....", and then proceed with the same narrative in line 182 and else where. Was baseline data collection part of this study? Please clarify confusion this because you statee in line 319 that "the study was conducted over a one year period". Please also clarify how you treated baseline and comparative data in this/these studies.

Your conclusion seems to "mute" other outcomes described and discussed in the study/effect of other intervention components aside from provision of a other-baby delivery pack. Please explain this.

Remove references from the conclusion section.

Line 343, "study findings also show that the intervention in implementable". What do you mean?? The authors did not investigate implementation and do not document any implementation outcomes in this particular study. They even mention quite a number of limitations including a lack of costing, cost-effectiveness data among others, making it premature to make such assertions/conclusions.

I would also consider the last six sentences of the conclusion to fall back into the discussion section, and authors to direct the readers on solid conclusions derived from the study data presented in this manuscript, plus make appropriate recommendations for further research/investigation.

7. PLOS authors have the option to publish the peer review history of their article (what does this mean?). If published, this will include your full peer review and any attached files.

Reviewer #1: No

Reviewer #2: No

Reviewer #3: No

---

## [Author Response · Author response to Decision Letter 1]

11 Sep 2023

Responses to Reviewers’ comments

Reviewer # 1: Minor revision:

Query 1: Line 61: R-studio is a development environment for R. It is important to state the version of R that was used for conducting the analysis. In fact, the use of R-studio is rarely noted in manuscripts.

Response: We appreciate the comment from the reviewer. We mentioned in the abstract and methods sections that the Analysis was conducted using R-studio statistical software version 4.2.1 (see pages and 7).

 Query 2: Line 59: Chi-square and independent t-tests are used to make comparisons between arms. Be sure to include the word "compare" or "comparisons" in this sentence.

Response: We appreciate the observation; we have made the change accordingly (page 2)

Query 3: Line 61: Clarify if "(p-value < 0.05)" implies that p-values < 0.05 were considered statistically significant.

Response: We appreciate the comment. We have now clarified that the p-value<0.05 was considered significant (see page 2 and 7). 

Query 4: Line 198: The independent t-test and chi-square tests are actually inferential tests rather than descriptive statistics. Descriptive statistics are means, standard deviations, frequencies, and percentages, etc.

Response: We thank the reviewer for the comment. We have now corrected the sentence to read: “Descriptive statistics (frequencies, percentages, proportions, means and standard deviations) was used to summarise participant socio-demographic and clinical data as well as institutional deliveries in intervention and control areas for each year” (see page 7)

Query 5: Supplementary Table 1 contains replicates of the variables.

Response: We have corrected the table and removed all the replicate variables (see table 1).

Query 6: Supplementary Tables 2 and 3: State a more precise p-values rather than > 0.05.

Response: The tables have been corrected accordingly (see tables 2 and 3)

Reviewer #2: General:

Query 1: The introduction is too brief and lacks key information for the reader. There is no information on ANC or on related interventions and their effectiveness. What have we already learned that led the authors to design and test this specific intervention? There should be sufficient information that a reader can glean this from the introduction.

Response: We thank the reviewer for the insightful feedback. We have edited the introduction and added a section on antenatal care [ see pages 4 and 5)

Query 2: This manuscript reports findings from a randomized trial. The CONSORT guidelines should be used to ensure all relevant information is included. I suggest the authors download the CONSORT checklist and verify all information is included.

Response: We appreciate the reviewer’s comment. The CONSORT guidelines was used and filled in accordingly

Query 3: The statistical analysis is not appropriate for the trial design. This was a cluster randomized trial; therefore, facility-level clustering needs to be accounted for in the analysis or standard errors will not be approrpriately estimated, potentially biasing the results and leading to incorrect interpretation. I have provided more details below. Moreover, characteristics that were different between arms at baseline should be adjusted for.

Response: We thank the reviewer for the comments. The respondent characteristics at baseline were compared between the intervention and control sites (see table 1). There was no significant difference between the two groups with regard to age, number of children and antenatal care booking. A significant difference between the two groups with regard to gravidity. Stratified analysis of variance ANOVA comparing the mean number of deliveries before and after the intervention in the same facilities accounts for any potential facility-level clustering. Moreover, we have explained that both intervention and control regions had similar health facilities regarding the location (rural), size and catchment population, socio-economic and demographic profiles. The two regions mainly served peasant and subsistence farmers of the same tribe who shared same cultural, traditional practices and beliefs. To ensure that the two regions were comparable with regard to population size, population data for the two regions was obtained from the Central Statistical Office in Zambia (see page 6). 

Query 4: There should be some analysis of participants lost to follow up. Are these women significantly different? How might this bias results? I cannot fully evaluate the results or discussion at this time given the limitations of the methods (lack of information). 

Response: We appreciate the comment from the reviewer. We mentioned that a total of 5,000 pregnant women were initially recruited into the study; meaning that they expressed interest and willingness to participate in the study. This was done at first contact in the antenatal care clinic. However, for various reasons, not clearly understood, 500 (10%) never returned to the health facility for follow up. Thus, they were deemed not part of the study since they were not exposed to the study and information on these is not available (see page 10). 

 Query 5: Were women not exposed to the intervention at different points of their pregnancy? The authors should expand on the timing of outcome measures relative to the intervention. If women were enrolled at their first ANC visit, it is unclear to me whether they were observed after the conclusion of the intervention as this is not clear from the methods section.

Response: We thank the reviewer for the insightful comment. Yes, women were identified from the ANC clinics, when they went for their first ANC visit, regardless of the stage of their pregnancy (gestation). The outcome of interest was coming to the health facility for delivery (institutional delivery). Thus, their follow up ended when they delivered; either at the health facility or home. They were not observed after the intervention. To ensure that no woman gave birth after the end of the follow up period, care was taken during recruitment to ensure that only women whose expected date of delivery (EDD) fell before 30th Nover, 2014 were recruited into the study. We picked on 30th November to allow for one month window for women who would experience prolonged gestation beyond 40 weeks to be observed. Participant recruitment ended in mid- year (June, 2014) so that the last delivery would be expected before the end of the intervention in December, 2013. In this case, no further follow up was not needed after the study. Moreover, this avoided the need for right censoring in our analysis (see page 8).

Abstract:

Query 6: Please define secondary outcomes reported as results here.

Response: We have made the correction accordingly (see page 2)

Query 7: The abstract attributes the difference in arms to the mother-baby pack, but the abstract notes that the intervention also included health education. Is the difference attributed to both or can the authors statistically isolate the effect of the mother-baby pack? Either way, this should be clear in the abstract

Response: We have corrected the conclusion section in the abstract to include. It now reads as follows: “These findings provide evidence for the effectiveness of the mother-baby delivery pack and additional health education sessions on increasing institutional deliveries in rural Zambia” (see page 2).

Introduction:

Query 8 : Line 82-83: “Home deliveries…” this sentence makes a claim that should be supported by a citation(s).

Response: We have included the citations accordingly (see page 4)

Query 9: Why is there no information or literature review on ANC in the introduction if this is a key component of the intervention evaluated? Please describe relevant background on ANC, including number of recommended visits and attendance rates (e.g., from the most recent DHS) in Zambia, and barriers to ANC (could be more broadly, as relevant to the study population).

Response: We thank the reviewer for the thoughtful comment. We have added information on ANC in Zambia (see page 4 and 5)

Methods:

Query 10: Are there any private facilities that can support deliveries?

Response: We have edited the section on the study setting and added more detail on the health facilities in the district. We have also mentioned that, at the time, there were no private health facilities providing obstetric and newborn care in the district (see page 6). 

Query 11: The secondary outcome measures need to be more clearly defined. For example, “knowledge of pregnancy danger signs” is vague.

Response: We have edited the sentence to read: “The secondary outcome measures were: 1) antenatal care service utilisation; 2) postnatal care service utilisation by mother and baby; 3) under-five clinic service utilisation

Query 12 : How was the randomization conducted?

Response: We used cluster sampling; allocation was done at the cluster/ rural region level; the district was stratified into two rural regions separated in the middle by the town centre. The region on the western side was allocated to the intervention arm; the one on the eastern side as the control arm. Eight health facilities were included in each arm (see page 6 under trial design and randomisation)

Query 13: Information about recruitment should be in the enrolment section (around line 167).

Response: This has been noted and corrected accordingly (see page 7)

Query 14: The abstract states analysis was done in R but the methods section states analysis was conducted in SPSS. Please clarify so that this is consistent.

Response: This was an error. We have now corrected it to read, “analysis was conducted using R-studio statistical software version 4.2.1” (see page 9).

Query 15: The information about data collection is insufficient. Please state where 2012 and 2013 data came from, and exactly what data was collected (e.g., precisely what measures were on the data extraction sheet, who extracted data). 

Response: We thank the reviewer for the insightful question and comment. We have now explained that both the baseline and intervention data were collected by a pair of data collectors under the supervision of the principal investigator. In order to establish baseline delivery data prior to commencement of the intervention, year-long delivery records for 2012 and 2013 were reviewed and delivery data collected from the delivery registers at each health facility in the study sites, using a data extraction sheet. The data extraction sheet comprised various sections including demographics (age, place of residence), gravidity, parity, gestation, pre-existing medical conditions (hypertension, HIV, anaemia), expected date of delivery, date of delivery, place of delivery, mode of delivery, delivery outcome, baby condition and outcome (see page 9). 

Query 16: What is the timing of paper-based questionnaires (“PNC” is far too vague, for example).

Response: We have explained that the paper-based questionnaire data was collected at 3 time points: at enrolment into the study during the ANC clinic, in the labour ward when the woman came for delivery and when the woman and baby came for the first PNC visit (see page 9)

Query 17: How were interviewers trained? Where were interviews conducted?

Response: We thank the reviewer for the comment. We have provided a detailed explanation on how the data collectors were trained: the training lasted 4 days (3 days three and 1 day practical). No interviews were conducted; rather a structured questionnaire was administered by midwives who were trained as data collectors. Above we have explained when and where the data was collected from (in the ANC, delivery and PNC clinics (see page 9)

Query 18: All non-descriptive analyses require clustering of the standard errors by facility given the study design. The authors should use regression models (e.g., linear, logistic depending on the outcome) with standard errors clustered by health facility. Otherwise, the independence assumption is violated which will bias standard errors and thus potentially lead to incorrect inferences about the statistical significance of the estimated relationships. Regression models estimating the effect of the intervention should also adjust for factors that were significantly differnet between arms at baseline, such as gravidity. Otherwise, this may be a spurious relationship.

Response: We thank the reviewer for the observation and comment. The aim of the tudy was to determine the effect of provision of additional health education during antenatal care (ANC) and a non-financial incentive on institutional deliveries. In our view, this effect can be tested using different models: analysis of various (ANOVA) or regression (linear, logistic or Poisson). In this case, we opted for one way ANOVA. We compared the mean number of deliveries between the intervention and control arms in the 3 years (2012, 2013 and 2014). Since there was no significant difference with regard to the variables of interest at baseline between the intervention and control arms, we attributed any observed difference in the number of deliveries to the intervention. Moreover, we believe that comparing the mean number of deliveries before and after the intervention in the same facilities accounted for any potential facility-level clustering (see response to query 3 above). In addition, we did not use linear regression because the aim of the study was not to identify the predictors of utilisation of delivery services. We believe the information we got and presented (F(1,46)=18.85, p<0.001) from ANOVA is as useful as the one we would have obtained from linear regression which looks at the goodness of fit for the model and the explained variance (R2). Due to other technical issues, we could not run analysis of covariance (ANCOVA) to account for the differences in gravidity. However, we have taken note of this and explained this weakness in our limitation section (see page 14)

Results

Query 19: Please include some analysis of participants lost to follow up. Are these women different from retained participants, and if so, how?

Response: We appreciate the comment; we have responded to this point already (see response to query 4 above).

Limitations:

Query 20: The authors should describe biases related to non-random allocation of facilities and whether there are potential selection biases related to including only public-sector facilities.

Response: We appreciate this guidance. We have described the biases related to non-random allocation of facilities (see page 14). We have explained that at the time of the study there were no private facilities providing obstetric and newborn healthcare services in the district (see page 10). 

Minor comments:

Query 21: Line 84: Demographic & Health Survey should be capitalized

Response: This has been corrected accordingly (see page 4)

Query 22: Line 108: change aim to objective, remove “study also” for clarity

Response: This has been corrected accordingly (see page 5)

Query 23: Lines 252 and 254: please include units

Response: This has been corrected accordingly (see page 11, 12)

Reviewer #3:

This is a very important and informative study. However, I have a few concerns that need to be addressed. 

Query 1: The statement "Participants in the intervention arm received ANC, health education and a mother-baby delivery pack when they arrived at the health facility for delivery" in your abstract is misleading. Please re-write.

Response: This has been corrected. The sentence now reads: In addition to the health education provided during routine ANC visits, participants in the intervention arm received health education and a mother-baby delivery pack when they arrived at the health facility for delivery (see page 2).

Query 2: Please re-write objectives stated in line 108-109 to align with the abstract and the rest of the document. 

Response: The objective has been corrected (see page 2)

Query 3: The document still has a lot of typos; e.g line 326-importance versus important; 327 "community intervention trials are the only appropriate study design suited...". Please revisit and re-write for clarity.

Response: We have proof-read the whole document and corrected all the typo errors. 

Query 4: You mention that study participants were identified from the first ANC visits, was this part of your screening criteria?

Response: Yes, the participants were identified during their first ANC visit so that they could be screened for their eligibility. Pregnant people who came for ANC at any gestation, regardless of the number of visits, was eligible to participate. However, since the outcome of interest was place of delivery (institutional or home delivery), care was taken to ensure that only women whose expected date of delivery (EDD) fell before 30th Nover, 2014 were recruited into the study (see page 7). 

Query 5: You later state these differences in health service utilization, alongside other variables like gravidity, first ANC utilization, how did you take care of these and other obvious confounders? Please describe your multivariate analysis in more detail.

Response: We thank the reviewer for the insightful comments. We have explained above (under reviewer # 2 query 18) that the objective of the study was to assess the effect of provision of additional health education during antenatal care (ANC) and a non-financial incentive on institutional deliveries. In our view, this effect can be tested using different models: analysis of various (ANOVA) or regression (linear, logistic or Poisson). In this case, we opted for one way ANOVA. We compared the mean number of deliveries between the intervention and control arms in the 3 years (2012, 2013 and 2014). 

Since there was no significant difference with regard to the variables of interest at baseline between the intervention and control arms, we attributed any observed difference in the number of deliveries to the intervention. Moreover, we believe that comparing the mean number of deliveries before and after the intervention in the same facilities accounted for any potential facility-level clustering (see response to query 3 above). In addition, we did not use multivariate linear regression models because the objective of the study was not to identify the predictors of utilisation of delivery services. We believe the information we got and presented (F(1,46)=18.85, p<0.001) from ANOVA is as useful as the one we would have obtained from linear regression which looks at the goodness of fit for the model and the explained variance (R2). 

Query 6: Please report ALL secondary outcomes in the outcome tables and results section before embarking on their discussion.

Response: We appreciate the guidance by the reviewer. We have now reported all the secondary outcomes in the results section (see page 13)

Query 7: Line 257 states, "health facility deliveries in the intervention and control sites from January 2012 to Dec 2014....", and then proceed with the same narrative in line 182 and else where. Was baseline data collection part of this study? Please clarify confusion this because you state in line 319 that "the study was conducted over a one year period". Please also clarify how you treated baseline and comparative data in this/these studies.

Response: We thank the reviewer for the comment. We have now explained that this was a three-year study (2012 to 2014) analysing baseline delivery data for 2012 and 2013 followed by one-year (1st January to 31st December 2014) prospective community intervention trial conducted in Monze district, Zambia (see page 6). 

Query 8: Your conclusion seems to "mute" other outcomes described and discussed in the study/effect of other intervention components aside from provision of a other-baby delivery pack. Please explain this.

Remove references from the conclusion section.

Response: The conclusion has been edited both in the abstract and main document and highlights all the outcomes described and discussed in the document.

Query 9: Line 343, "study findings also show that the intervention in implementable". What do you mean?? The authors did not investigate implementation and do not document any implementation outcomes in this particular study. They even mention quite a number of limitations including a lack of costing, cost-effectiveness data among others, making it premature to make such assertions/conclusions. I would also consider the last six sentences of the conclusion to fall back into the discussion section, and authors to direct the readers on solid conclusions derived from the study data presented in this manuscript, plus make appropriate recommendations for further research/investigation.

Response: This section has been edited and the sentence, “study findings also show that the intervention is implementable” has been deleted. We have also made recommendations for future research/investigation (see page 16)

---

## [Decision Letter · Decision Letter 2]

7 Nov 2023

PONE-D-23-04893R2Effect of a non-financial incentive on institutional deliveries: A community intervention trial to address maternal mortality in rural ZambiaPLOS ONE

Dear Dr. Sialubanje,

Thank you for submitting your manuscript to PLOS ONE. After careful consideration, we feel that it has merit but does not fully meet PLOS ONE’s publication criteria as it currently stands. Therefore, we invite you to submit a revised version of the manuscript that addresses the points raised during the review process.

We look forward to receiving your revised manuscript.

Kind regards,

Hector Lamadrid-Figueroa, MD, ScD

Academic Editor

PLOS ONE

Journal Requirements:

Additional Editor Comments:

Your manuscript is close to acceptance. Please address issues raised by reviewers 2 & 3.

Reviewers' comments:

Reviewer's Responses to Questions

**Comments to the Author**

1. If the authors have adequately addressed your comments raised in a previous round of review and you feel that this manuscript is now acceptable for publication, you may indicate that here to bypass the “Comments to the Author” section, enter your conflict of interest statement in the “Confidential to Editor” section, and submit your "Accept" recommendation.

Reviewer #1: All comments have been addressed

Reviewer #2: (No Response)

Reviewer #3: All comments have been addressed

2. Is the manuscript technically sound, and do the data support the conclusions?

Reviewer #1: (No Response)

Reviewer #2: Yes

Reviewer #3: Yes

3. Has the statistical analysis been performed appropriately and rigorously? 

Reviewer #1: (No Response)

Reviewer #2: Yes

Reviewer #3: Yes

4. Have the authors made all data underlying the findings in their manuscript fully available?

Reviewer #1: (No Response)

Reviewer #2: Yes

Reviewer #3: Yes

5. Is the manuscript presented in an intelligible fashion and written in standard English?

Reviewer #1: (No Response)

Reviewer #2: Yes

Reviewer #3: Yes

6. Review Comments to the Author

Reviewer #1: (No Response)

Reviewer #2: The authors have thoughtfully addressed many of the points I raised in my first review of this paper. I appreciate their careful consideration of these broad issue, which required a significant revision on their part. This version of the manuscript is greatly improved. Overall, this version was much easier to read, and I especially appreciated the expanded introduction and methods sections.

It might be underselling the importance of this trial to refer to the mother-baby pack as a non-financial incentive. This is a key cost-related barrier that the authors directly addressed and I would hate for this finding to get lost, which it somewhat does in the current abstract and first paragraph of the discussion, and the conclusion. The authors might add that the pack in both places that the pack includes essential delivery supplies, which would otherwise be an out-of-pocket expense. Adding a few sentences to make it clear that an aim of the trial is to test the effect of covering this potential expense in-kind that is a major barrier to utilizing facility delivery may increase the reach of these important findings.

In general this version is much clearer and follows CONSORT guidelines for reporting trial results. I commend the authors for this very significant revision. My comments below are targeted towards further improving the manuscript.

Abstract

- It might be helpful to note that the 5,000 women were identified as eligible or enrolled at baseline but were not exposed to the intervention; otherwise this raises questions about loss to follow up (this point was addressed in the manuscript and response to reviewers). I would also clarify here that the 12.4% who completed the study were not included in analysis because of incomplete data (otherwise it is unclear why they were not included.

- The secondary outcomes remain poorly defined. For example, does “ANC” mean obtained any ANC vs. initiated ANC in the first trimester vs. completed at least 4 ANC visits, or something else?

- If the authors are concerned about word limit in the abstract given these other suggestions, they could remove the statistical software as this is not typically necessary to report in the abstract (the main text is sufficient).

Introduction:

- If 33% of women are delivering at home, I would not say it is consistent to state “most women still give birth at home” (line 104). Please revise “most” (of course this is a not insubstantial proportion and I support the authors’ efforts to increase facility delivery). It does seem that at study baseline a majority of women in the district had home deliveries but this is not supported by the ZDHS data. If the authors have access to the 2013 ZDHS, they might estimate the proportion of home deliveries in rural areas of the study region, which would be available in the dataset and might be >50%.

Methods:

- Intervention: Could the authors clarify, is it that mother-baby packs were made available for free at all facilities in the intervention area, or that each woman was specifically given one (say, at her home) around the time of delivery? The mother-baby pack should be stated as part of the intervention in the first sentence of this paragraph.

- Randomization: how was the actual randomization done, was it a coin flip or something else? Or did someone decide that west would be intervention?

- Outcomes: same comment as in the abstract, the secondary outcomes remain poorly defined. Please specify how each outcome is operationalized, e.g., “mother received a postnatal care visit within 48 hours of delivery.” This is somewhat done in the results but should be here as well.

- If women are followed through their first PNC visit, does this mean that under 5 clinic utilization refers to PNC for the baby/newborn follow up? It might be helpful to be more explicit about this, as I assumed this was care for acute illnesses until I read the results section.

Results:

- I appreciate that the authors included all p-values. However, while we might state that a p-value of .07 is not significant, it does suggest that there is an underlying difference that may be meaningful in some way or may be significant if the sample size was a little larger (line 299). The authors

Discussion:

- Paragraph beginning on line 351: what were the non-financial incentives in the other studies referenced? It would be helpful to more directly compare what kinds of incentives they were.

- This study does have strengths as the authors note on the paragraph starting on line 409. However, I think it not reasonable to say that this was the only possible study design for this particular trial because in theory mother-baby packs and health education could be randomized at the individual level (I agree with the authors that theirs is an appropriate design). I would also remove the sentence starting on line 417; yes longer data collection periods may have challenges but that does not mean that the accuracy is always lower.

Minor comments:

- Line 158: “assent” instead of “ascent”

Reviewer #3: Description of secondary outcomes is still vague

How did you choose the eight health centres?

You state that the questionaires were administered by the midwives who were trained as data collectors. Are these the same midwives running the labour ward, AND, PNC clinics? How did you deal with potential bias during data collection?

At what point was PNC data collected?

7. PLOS authors have the option to publish the peer review history of their article (what does this mean?). If published, this will include your full peer review and any attached files.

Reviewer #1: No

Reviewer #2: No

Reviewer #3: No

---

## [Author Response · Author response to Decision Letter 2]

15 Nov 2023

Response to reviewer comments

Reviewer # 2

Query 1: The authors have thoughtfully addressed many of the points I raised in my first review of this paper. I appreciate their careful consideration of these broad issue, which required a significant revision on their part. This version of the manuscript is greatly improved. Overall, this version was much easier to read, and I especially appreciated the expanded introduction and methods sections.

 It might be underselling the importance of this trial to refer to the mother-baby pack as a non-financial incentive. This is a key cost-related barrier that the authors directly addressed and I would hate for this finding to get lost, which it somewhat does in the current abstract and first paragraph of the discussion, and the conclusion. The authors might add that the pack in both places that the pack includes essential delivery supplies, which would otherwise be an out-of-pocket expense. Adding a few sentences to make it clear that an aim of the trial is to test the effect of covering this potential expense in-kind that is a major barrier to utilizing facility delivery may increase the reach of these important findings

Response: We appreciate the complements from the reviewer and the concern about our referring to the mother-baby pack as a non-financial incentive. We have now corrected this both in the abstract and discussion. We have also corrected the title and conclusion sections accordingly. The aim now reads as: To test the effect of provision of additional health education during antenatal care (ANC) and a mother-baby delivery pack on institutional deliveries in Monze, Zambia.

Abstract

Query 2: It might be helpful to note that the 5,000 women were identified as eligible or enrolled at baseline but were not exposed to the intervention; otherwise this raises questions about loss to follow up (this point was addressed in the manuscript and response to reviewers). I would also clarify here that the 12.4% who completed the study were not included in analysis because of incomplete data (otherwise it is unclear why they were not included.

Response: We thank the reviewer for this observation. We have made the correction accordingly (page 2 line 31 to 34)

Query 2: The secondary outcomes remain poorly defined. For example, does “ANC” mean obtained any ANC vs. initiated ANC in the first trimester vs. completed at least 4 ANC visits, or something else?

- If the authors are concerned about word limit in the abstract given these other suggestions, they could remove the statistical software as this is not typically necessary to report in the abstract (the main text is sufficient).

Response: We appreciate the comment by the reviewer. We have now explained that ANC, PNC and under-five service utilisation means the time of the first visit, number of visits completed according to the national guidelines (see page 8).

A detailed explanation about ANC and PNC services in the country is provided in the introduction section.

Introduction:

Query 3: If 33% of women are delivering at home, I would not say it is consistent to state “most women still give birth at home” (line 104). Please revise “most” (of course this is a not insubstantial proportion and I support the authors’ efforts to increase facility delivery). It does seem that at study baseline a majority of women in the district had home deliveries but this is not supported by the ZDHS data. If the authors have access to the 2013 ZDHS, they might estimate the proportion of home deliveries in rural areas of the study region, which would be available in the dataset and might be >50%.

Response: We appreciate the guidance from the reviewer. We have corrected the sentence accordingly (see page 5 lines 122 to 124). We have also provided the statistics for the study district in the methods section under study setting (see page 7 lines 180 to 181)

Methods:

Query 4: Intervention: Could the authors clarify, is it that mother-baby packs were made available for free at all facilities in the intervention area, or that each woman was specifically given one (say, at her home) around the time of delivery? The mother-baby pack should be stated as part of the intervention in the first sentence of this paragraph.

Response: We have clarified that the mother-baby delivery packs were kept at the health facility; pregnant women only received them at the time of delivery if they went to deliver at the health facility. Information about the packs and their content was provided during the health education sessions as women went for their ANC visits (see pages 8 and 9). We have corrected the first sentence accordingly. 

Query 5: Randomization: how was the actual randomization done, was it a coin flip or something else? Or did someone decide that west would be intervention?

Response: We appreciate the comment from the reviewer. We have now explained in detail how randomisation was done (see page 6 and 7)

Query 6: Outcomes: same comment as in the abstract, the secondary outcomes remain poorly defined. Please specify how each outcome is operationalized, e.g., “mother received a postnatal care visit within 48 hours of delivery.” This is somewhat done in the results but should be here as well. If women are followed through their first PNC visit, does this mean that under 5 clinic utilization refers to PNC for the baby/newborn follow up? It might be helpful to be more explicit about this, as I assumed this was care for acute illnesses until I read the results section

Response: We have now explained how each secondary outcome was operationalised (see page 9). For PNC there were three measures: did the mother and her baby receive PNC? Time after delivery when PNC was received? Number of PNC visits. Similarly, for ANC, the measures were: did the pregnant woman use ANC? Gestation at ANC booking, and number of ANC visits completed.

Results:

Query 7: I appreciate that the authors included all p-values. However, while we might state that a p-value of .07 is not significant, it does suggest that there is an underlying difference that may be meaningful in some way or may be significant if the sample size was a little larger (line 299). 

Response: We appreciate the observation by the reviewer. We have now corrected this sentence to read: “There was a notable difference in the number of children between the respondents from the intervention (mean=3.6, SD=2.3) and control arms (mean=3.4, SD=2.2); however, the difference did not reach statistical significance (p=0.07) (see page 31)

Discussion:

Query 8: Paragraph beginning on line 351: what were the non-financial incentives in the other studies referenced? It would be helpful to more directly compare what kinds of incentives they were.

Response: We appreciate the guidance from the reviewer. We have now described the incentives provided in the cited studies (see page 15 lines 397 to 407)

Query 9: This study does have strengths as the authors note on the paragraph starting on line 409. However, I think it is not reasonable to say that this was the only possible study design for this particular trial because in theory mother-baby packs and health education could be randomized at the individual level (I agree with the authors that theirs is an appropriate design). I would also remove the sentence starting on line 417; yes longer data collection periods may have challenges but that does not mean that the accuracy is always lower.

Response: We appreciate the concern from the reviewer. We have now deleted the sentence accordingly (see page 17 lines 466 and 467)

Minor comments:

Query 10: Line 158: “assent” instead of “ascent”

Response: We have corrected this and all other typo errors in the document

Reviewer #3: 

Query 1: Description of secondary outcomes is still vague

Response: we appreciate the observation by the reviewer. We have now described the secondary outcomes in detail. We have explained how each secondary outcome was operationalised (see page 9). For PNC there were three measures: did the mother and her baby receive PNC? Time after delivery when PNC was received? Number of PNC visits. Similarly, for ANC, the measures were: did the pregnant woman use ANC? Gestation at ANC booking, and number of ANC visits completed.

Query 2: How did you choose the eight health centres?

Response: We appreciate the observation from the reviewer. We have now explained how the 8 health centres were selected (see page 6 and 7)

Query 3: You state that the questionaires were administered by the midwives who were trained as data collectors. Are these the same midwives running the labour ward, AND, PNC clinics? How did you deal with potential bias during data collection?

Response: We thank the reviewer for the observation. We have now explained that the midwives who collected the data were not involved in the service (ANC, delivery and PNC) provision. We have also mentioned that several measures were taken to ensure quality and avoid bias during the data collection process: data collectors were trained and worked under the supervision of the principla investigators and other research team members. In addition, midwives who participated in data collection were not involved in service provision (see page 10). 

Query 4: At what point was PNC data collected?

Response: We thank the reviewer for the question. We mentioned under data collection section that PNC data was collected after delivery and when women came for PNC visits (see page 10)

---

## [Editor Report · Decision Letter 3]

5 Dec 2023

Effect of a mother-baby delivery pack on institutional deliveries: A community intervention trial to address maternal mortality in rural Zambia

PONE-D-23-04893R3

Dear Dr. Sialubanje,

We’re pleased to inform you that your manuscript has been judged scientifically suitable for publication and will be formally accepted for publication once it meets all outstanding technical requirements.

Kind regards,

Hector Lamadrid-Figueroa, MD, ScD

Academic Editor

PLOS ONE

Additional Editor Comments (optional):

Congratulations on your fine work!
---

## [Editor Report · Acceptance letter]

14 Dec 2023

PONE-D-23-04893R3 

PLOS ONE

Dear Dr. Sialubanje, 

I'm pleased to inform you that your manuscript has been deemed suitable for publication in PLOS ONE. Congratulations! Your manuscript is now being handed over to our production team.

Kind regards, 

on behalf of

Dr. Hector Lamadrid-Figueroa 

Academic Editor

PLOS ONE